# Sensory-Adapted Dental Environment for the Treatment of Patients with Autism Spectrum Disorder

**DOI:** 10.3390/children9030393

**Published:** 2022-03-10

**Authors:** Antonio Fallea, Rosa Zuccarello, Michele Roccella, Giuseppe Quatrosi, Serena Donadio, Luigi Vetri, Francesco Calì

**Affiliations:** 1Oasi Research Institute-IRCCS, Via Conte Ruggero 73, 94018 Troina, Italy; afallea@oasi.en.it (A.F.); rzuccarello@oasi.en.it (R.Z.); cali@oasi.en.it (F.C.); 2Department of Psychology, Educational Science and Human Movement, University of Palermo, 90128 Palermo, Italy; michele.roccella@unipa.it; 3Department of Sciences for Health Promotion and Mother and Child Care “G. D’Alessandro”, University of Palermo, 90128 Palermo, Italy; giuseppe.quatrosi01@community.unipa.it (G.Q.); serenadonadio@libero.it (S.D.)

**Keywords:** sensory processing, dental anxiety, autism

## Abstract

Purpose: The importance of dental care and oral hygiene is often underestimated in people with autism spectrum disorder (ASD). Comorbidity with dental anxiety is greater in ASD subjects who also show unusual reactions to sensory stimuli. The aim of our study was to assess the efficacy for a sensory-adapted environment and targeted methods in reducing anxiety and positively influencing cooperation in children with ASD during a dental examination or specific treatments. Material and methods: The sample consisted of 50 Italian children with a diagnosis of ASD (36 males and 14 females; aged 9–10 years) presenting with mild intellectual disability (ID) and verbal language skills. The subjects enrolled in the study had at least two decayed teeth and all were treated in two different dental environments: regular dental environment (RDE) and sensory-adapted dental environment (SADE). Results: 20% of the sample was successfully treated in RDE, while 68% of subjects were successfully treated in SADE. Conclusions: Results suggest that a sensory-adapted environment positively affects the therapeutic dental treatment in patients with ASD and reaffirm that sensory dysregulation in children with ASD is a crucial factor influencing the successful outcome of oral care.

## 1. Introduction

Dental care and oral health are very important and are a crucial aspect in the overall health and quality of life for individuals. For some time now, the World Health Organization has stressed the importance of oral health among the health topics that each Member State should pursue. This topic, however, does not attract a great deal of attention and is often disregarded and undervalued in people with autism spectrum disorder (ASD) who present with abnormalities in communication, socialization, and restricted-repetitive behavioral patterns. Such difficulties, occurring in children from their earliest years of life and in different ways, are classified according to the severity of symptoms and are often associated with intellectual disability (ID). Moreover, these conditions may be associated with varying degrees of cognitive disability [1]. People with ASD have impaired prediction skills and lack flexibility in expectations, which are essential elements during a dental examination [2]. Children with ASD are often unable to tolerate dental interventions because of fear associated with sights and sounds in the dental setting (perceptual hypersensitivity). Furthermore, a greater dental anxiety and unusual reactions to sensory stimuli are observed in non-collaborative patients presenting with ID, as reported by Fallea et al. in 2016 [3]. Therefore, general anesthesia is sometimes required for regular dental procedures, which could lead to complications [4]. In a study carried out by Cermak et al. [5], the authors analyzed the behavioral distress, pain, and sensory discomfort both in children with autism and in typically developing children during a dental treatment performed in a regular dental environment (RDE) and in a sensory-adapted dental environment (SADE). Results showed positive benefits from interventions carried out in an adapted environment [5]. In 2007, Shapiro et al. [6] argued that a SADE might be effective in reducing anxiety and inducing relaxation. In their study, they achieved better results by using a room with lighting effects and vibro-acoustic stimuli than those obtained with a treatment performed in a normal environment. In 2009, Shapiro et al. [7] investigated the influence of an adapted dental environment on anxiety in children with developmental disabilities compared to their typically developing peers. In both groups, performance was more effective in the adapted environment than in the usual setting; in addition, the difference between the two environments was more evident in children with developmental disabilities. Behavioral and physiological measures of stress and anxiety were collected by Stein et al. [8] during dental clearings in a sample of 44 children (22 typical and 22 ASD). Results showed that children with ASD showed greater distress and anxiety compared to the typical group. In 2015, Nelson et al. [9] conducted a literature review to obtain relevant information which could be useful for proper dental care in children with ASD. They identified some educational principles to be successfully used and applied in the dental setting: social stories, video modeling, dividing dental treatment into sequential components, and modification of the environment to minimize sensory triggers. A study by Bartolomè-Villar et al. [10] aimed to identify and analyze the existing literature on the oral conditions of children with ASD and children with sensory deficits compared to healthy children. A total of 10 articles were found for ASD and six for sensory.

In a recent systematic review, Ismail et al. analyzed four studies assessing the effectiveness of SADE on children with special needs who received dental treatment. The studies analyzed showed that children with special needs treated in SADE make significant improvements in terms of physiological changes, behavior, pain, and sensory discomfort [11].

In light of the aforementioned studies, the aim of the present study was to assess the efficacy for a sensory-adapted environment and targeted methods in reducing anxiety and positively influencing children with ASD to cooperate during a dental examination or specific treatments. Undergoing dental treatment is a challenge for children’s self-control and self-modulation skills. Children with ASD also adapt with greater difficulty to a normal dental setting because of their sensory anomalies. Hypo and hypersensitivity and the difficulty in discriminating between different sensorial channels can worsen the stress and anxiety perceived by ASD children when visiting a dentist, thus leading inevitably to a poor final outcome in the dental procedure. Therefore, the purpose of this study was to check whether a sensory-adapted dental environment may increase the percentage of ASD patients successfully treated for their class-1 caries.

## 2. Materials and Methods

### 2.1. Sample

The study was approved by the local Ethical Committee of the Oasi Research Institute–IRCCS in Troina, Italy (CE17/06/2013 OASI). Written informed consent was obtained from the participants. The subjects were recruited among 55 patients referred to diagnostic and rehabilitation services at the Oasi Research Institute–IRCCS (Troina, Italy), a research institute dealing with treatment and rehabilitation in the field of intellectual disabilities.

The inclusion criteria were: (a) a stable diagnosis of ASD made by expert qualified psychologists [12] following the DSM-5 criteria and after the administration of the Childhood Autism Rating Scale (C.A.R.S.) [13], Autism Diagnostic Observation Schedule (ADOS) [14], and Autism Diagnostic Interview—Revised (ADI-R) [15]; (b) having at least two permanent teeth with caries in class I; and (c) absence of comorbid psychiatric or neurological pathologies.

Exclusion criteria were: (a) patients with one permanent tooth with class-I caries or no permanent teeth with caries.

Starting from 55 patients initially recruited, the sample was reduced to 50 patients because five of them had only one decayed tooth (exclusion criterion). Indeed, according to the study design, all patients should present with at least two decayed permanent teeth. In a first phase, they were brought to the RDE in order to try and treat one decayed permanent tooth and, subsequently, all patients were led to SADE to treat the second decayed permanent tooth. If we had recruited patients with only one decayed tooth, we would not have had the possibility in the second phase to be able to cure the second decayed tooth.

### 2.2. Procedures: Environment

In our study, we used two different dental environments:-A regular dental environment (RDE);-A sensory-adapted dental environment (SADE).

The same dentist, with 20 years’ experience in the treatment of children with ASD, treated the two permanent teeth with class-I caries in the different dental environments. The two different dental environments are located in the Odontostomatology Unit of Oasi Research Institute–IRCCS in Troina.

In contrast to the RDE, the SADE was provided with a screen to project movies, cartoons, or advertisements (already known by the subject and previously provided by the family), soft lighting, and a sponge-coated dental turbine drill to minimize the noise. Firstly, all patients were treated in the RDE and, after two months, dental interventions were performed in the SADE.

In the first phase in the RDE, the patient was seated on the dental chair and through the dental turbine, the class-1 cavity was removed and the filling was made with glass ionomer restorative material.

In the second phase, the patient was transferred to the SADE with soft lighting. The patient was then seated on the dental chair while a screen projected movies, cartoons, or advertisements (already known by the subject and previously provided by the family). By using a sponge-coated dental turbine drill to minimize the noise, the class-1 cavity was removed and, once removed, the resulting cavity was filled with glass ionomer restorative material (Figure 1).

The dental treatments for 50 patients were scheduled one per day; therefore, the average time elapsed between the RDE and the SADE treatment was about two months.

The study was carried out between January 2014 and April 2014. According to our a priori hypotheses, the adapted setting would have led to better results.

### 2.3. Statistical Analyses

The experimental design evaluated the differences between “before” and “after” by using a 2 × 2 contingency table McNemartest (statistical procedures were performed with SPSS—version 26.0.0.0). Since the total *N* for a 2 × 2 chi-squared table is less than about 40, the Yates continuity correction was used. P values less than 0.05 were considered to be statistically significant. The statistical appropriateness of the chi-squared tests used in this study was assessed “a posteriori” by first calculating the effect size and then calculating the corresponding sample size required with α = 0.05, resulting in an actual power of 98% and a required sample size of 45 for the 2 × 2 matrix.

## 3. Results

The final sample for the study consisted of 50 patients (14 females and 36 males aged 9 to 10 years), each with two permanent teeth with class 1-caries. All subjects received a stable diagnosis of ASD by expert qualified psychologists according to the DSM-5 criteria and after administration of C.A.R.S., ADOS, and ADI-R; moreover, they presented with mild ID (IQ ranging from 50 to 70) and verbal language skills in the absence of concomitant psychiatric and neurological pathologies.

In the first phase of the study, all patients of the sample were taken in RDE. In this phase, 11 patients (4 females and 7 males) were positively treated for class-1 caries of a permanent tooth; in the remaining 39 patients (10 females and 29 males), this was not possible. In the second phase, the entire sample were led to SADE. During the SADE phase, 34 patients (9 females and 25 males) were positively treated for class-1 caries of the second permanent tooth, while in 16 patients (5 females and 11 males) it was not possible to treat the class-1 caries. (Table 1 and Figure 2). Furthermore, it should be highlighted that the 11 patients positively treated in RDE received a second positive treatment for the class-1 caries of the second permanent tooth in SADE.

Difference between the two groups (RDE vs. SADE) is statistically significant at the 0.05 level (chi-squared test = 21.043 with 1 degree of freedom, *p*-value < 0.0001). The duration of each treatment was not detected.

Statistical analyses were conducted to understand the statistical significance between RDE and SADE in females and males respectively. The results show a statistical significance for male patients (chi-squared test = 18.225, df = 1, *p*-value = 0.00002; Yates’s chi-squared test = 16.256, df = 1, *p*-value = 0.000055) and not for female patients (chi-squared test = 3.5897, df = 1, *p*-value is 0.058137; Yates’s chi-squared test = 2.297, df = 1, *p*-value 0.12958781).

## 4. Discussion

Children with ASD have poor oral hygiene resulting from difficulties in tolerating home and professional oral care; this is mainly due to differences in sensory processing, uncooperative behaviors, and difficulties in finding and accessing professional oral hygiene services [16].

Interestingly, our result shows a statistical significance for male patients and not for female patients. This difference can be explained by sex differences in sensory processing in ASD children [17] or by the small sample size of ASD female children. We hope that future studies with a greater sample size could shed light on this aspect.

The present study has several limitations, and the results must be interpreted with caution: (a) the sample examined was small and consisted mainly of males, (b) there was no control sample of children with typical development, (c) the evaluation in the effectiveness of the dental treatment was left exclusively to the dentist’s judgment.

Moreover, all children in the sample were brought to a traditional dental environment to try and treat the class-1 caries, and at a later time they were brought to a sensory-adapted dental environment; therefore, in the second phase of the study, ASD children had already experienced the dental environment. This first phase, even if performed in a regular dental environment, may have favored a better compliance during the second experience regardless of the environment used. However, we believe that ASD individuals tend to crave sameness, thus a single experience is not enough to produce relevant habituation phenomena.

Consistent with previous literature findings, the results from our study highlight that an adapted environment positively affects the therapeutic dental treatment performed on patients with ASD. Interestingly, the use of a SADE leads to significant successful treatment of caries in patients with ASD.

For many children with ASD, dental care represents one of the main stressors, so parents tend to avoid dental examinations for their children, thus causing irreversible and irreparable damages over time.

In a randomized crossover study performed by Cermak et al., 22 children with ASD and 22 typically developing children received two dental cleanings. One treatment was carried out in an RDE and the other was carried out in a SADE. Their results were similar to our study, showing positive benefits from interventions carried out in an adapted environment and patients, families, dentists, clinic staff, and investigators positively responded to the SADE experience [18].

Watching a video serves as a distractor to efficiently reduce the anxiety associated with a non-collaborative behavior presented by a child with ASD. The use of soft lighting in a sensory-adapted environment minimizes the disruption from visual overstimulation [6,19,20]. The noise reduction in the dental drill decreases anxiety and fear in the patient [21]. The association among these three environmental changes tripled the successful treatment of patients’ caries, leading to a positive impact on the cooperation from children with ASD during dental treatments. What seems to mainly affect the success of dental treatment is the degree of a patient’s cooperation. The lack of cooperation during dental sessions could also be linked to previous negative experiences to which the patient has been exposed in a standard dental environment. We also believe that a specifically trained team of professionals (dentists, hygienists, nurses, pedagogue, and other specialists) able to interact together, can perform high-quality treatments in the shortest possible time while supporting the patient and his/her family [22]. Awareness of the patient’s medical and dental problems is essential, together with the ability to identify and manage the peculiar behavioral dynamic processes occurring in each individual and family.

In this context, it is important to also emphasize the role of the waiting room. A sensory-adapted waiting room environment seems to be less important in reducing the anxiety in typically developing children compared to other parameters, such as longer waiting time prior to treatment and visit purpose [23].

Similarly, an appropriate design of dental waiting areas improves the waiting experience and reduces the preoperative anxiety before a dental appointment. For instance, the majority of children prefer waiting in a room with natural light, walls with pictures or posters, looking at an aquarium or watching television, listening to music, and the possibility to play [24,25].

Consistent with this evidence, at Oasi Research Institute, we chose not to keep children with ASD waiting in a traditional waiting room, but let them wait their turn in the playroom or in recreational spaces.

## 5. Conclusions

Our results, in line with evidence in the scientific literature, suggest that a sensory-adapted environment positively affects the therapeutic dental treatment in male patients with ASD and reaffirm that sensory dysregulation in children with ASD is a crucial factor influencing the successful outcome in oral care. Future research studies, as well as the development and implementation of further environmental changes, are needed to obtain successful dental treatments for people with ASD with the aim of providing an experience with a greater perception of comfort, a strength in this particular population.

Although parents of children with ASD tend to consider oral health less important than the primary disease, more attention is required for their oral healthcare through targeted approaches to improve the wellbeing of individuals and their families.

## Figures and Tables

**Figure 1 children-09-00393-f001:**
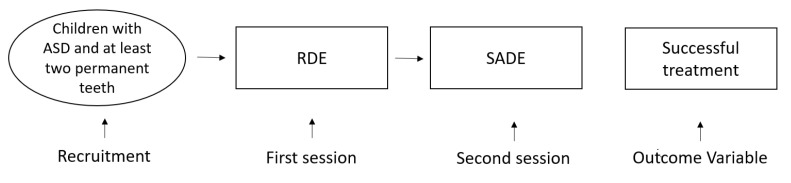
Overview of study design. Regular dental environment (RDE); sensory-adapted dental environment (SADE).

**Figure 2 children-09-00393-f002:**
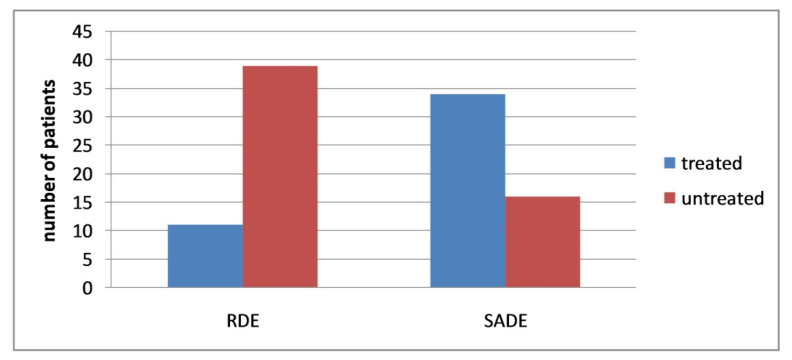
Therapeutic treatment in a regular dental environment (RDE) and an adapted dental environment (SADE) in overall (males and females) patients with ASD. (*N* = 50).

**Table 1 children-09-00393-t001:** Difference by sex in treatment success in regular dental environment (RDE) and in sensory-adapted dental environment (SADE).

RDE	Male	%	Female	%
Treated	7	19.44	4	28.57
Untreated	29	80.56	10	71.43
Tot	36	100	14	100
**SADE**	**Male**	**%**	**Female**	**%**
Treated	25	69.44	9	64.29
Untreated	11	30.56	5	35.71
Tot	36	100	14	100

## Data Availability

The data that support the findings of this study are available from the corresponding author upon reasonable request.

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
