# Peer review of "Sensory-Adapted Dental Environment for the Treatment of Patients with Autism Spectrum Disorder"

_children, 2022, doi:10.3390/children9030393_

Round 1

Reviewer 1 Report

The paper has to be improved language-wise a lot more, and I have noticed a couple of grammatical errors.

There were no details on the provision of treatment in both the appointments.

Since the authors mentioned inclusion  criteria, the below statements 

The inclusion criteria were: a) a stable diagnosis of ASD made by expert qualified psychologists [12] following the DSM-5 criteria and after the administration of Childhood Autism Rating Scale (C.A.R.S.) [13], Autism Diagnostic Observation Schedule (ADOS) [14] and Autism Diagnostic Interview - Revised (ADI-R) [15]; b) having at least two permanent teeth with caries in Class I; c) absence of comorbid psychiatric or neurological pathologies

The details on the representative sample were missing in the results section.

Please provide the details on treatments. 

RDE-11 patients and SADE- 34 patients were treated; the data on this context was missing.

I would be interested in seeing the data collection form as a supplemental file.

Discussion only based on RDE and SADE if authors could able to state the treatments, then it will help establish further prospects.

The present form of discussion was not really focused on the object of the study. 

I also recommend the authors draw a flow chart that helps understand the study design.

The below queries also not been clearly stated:

1. Methodology: there is confusion in subject selection kindly rephrase the statement.

2. Need explanation on sample size. "Since dental decay, at least two teeth was a mandatory inclusion criterion, five subjects were excluded from the study; thus, the final sample consisted of 50 patients, 14 females, and 36 males." Comprehensive details are required on inclusion and exclusion criteria.

3. 

Not very clear. Use the below citations for comparison with your findings.:

Fux-Noy A, Zohar M, Herzog K, Shmueli A, Halperson E, Moskovitz M, Ram D. The effect of the waiting room's environment on level of anxiety experienced by children prior to dental treatment: a case control study. BMC Oral Health. 2019 Dec 30;19(1):294.

Moana-Filho EJ, Alonso AA, Kapos FP, Leon-Salazar V, Durand SH, Hodges JS, Nixdorf DR. Multifactorial assessment of measurement errors affecting intraoral quantitative sensory testing reliability. Scand J Pain. 2017 Jul;16:93-98.

Panda A, Garg I, Shah M. Children's preferences concerning ambiance of dental waiting rooms. Eur Arch Paediatr Dent. 2015 Feb;16(1):27-33.

Cermak SA, Stein Duker LI, Williams ME, Dawson ME, Lane CJ, Polido JC. Sensory Adapted Dental Environments to Enhance Oral Care for Children with Autism Spectrum Disorders: A Randomized Controlled Pilot Study. J Autism Dev Disord. 2015;45(9):2876-2888. doi:10.1007/s10803-015-2450-5

Author Response

Dear Reviewers,

I would like to thank you for your valued comments and suggestions to the article. As requested, we made all the necessary changes in our manuscript to address your concerns which you can find detailed described below. The main changes are written in red in the manuscript. Thus, according to the changes made in the revised manuscript and the responses provided below, I hope you will consider the manuscript suitable for publication. If there are any further questions, please feel free to contact me.

Sincerely,

Luigi Vetri,

ORCID iD 0000-0002-0121-3396

Oasi Research Institute-IRCCS, Troina, Italy

Address: Via Conte Ruggero 73 - 94018 TROINA (EN);

Phone: +390935936809

Email: [email protected][email protected]

Reviewer 1

The paper has to be improved language-wise a lot more, and I have noticed a couple of grammatical errors.

Answer: Many thanks for your valued suggestions. If the article will be accepted we will use the MDPI service of linguistic revision.

There were no details on the provision of treatment in both the appointments.

Answer: Thanks for the suggestion. The only treatment performed in the study was the treating of the class-1 caries. In the first phase in RDE the patient was seated on the dental chair and through the dental turbine the class-1 caries was removed and the filling was made with glass ionomer restorative material.

In the second phase, the patient was transferred to the SADE with soft lighting. The patient was then seated on the dental chair while a screen was projecting movies, cartoons or advertisements (already known by the subject and previously provided by the family). By using a  sponge-coated dental turbine drill to minimize the noise, the class-1 caries was removed and, once removed, the resulting cavity was filled with glass ionomer restorative material.

These details have been added in Materials and methods sections (2.2. Procedures: Environment), moreover a graph showing the study design have been also added (see figure 1).

Since the authors mentioned inclusion  criteria, the below statements 

The inclusion criteria were: a) a stable diagnosis of ASD made by expert qualified psychologists [12] following the DSM-5 criteria and after the administration of Childhood Autism Rating Scale (C.A.R.S.) [13], Autism Diagnostic Observation Schedule (ADOS) [14] and Autism Diagnostic Interview - Revised (ADI-R) [15]; b) having at least two permanent teeth with caries in Class I; c) absence of comorbid psychiatric or neurological pathologies. The details on the representative sample were missing in the results section.

Answer: Thanks for the suggestion. More details about the samples have been added in the result section as requested. Please, consider that ours is a non-probability purposive sampling.

Please provide the details on treatments. 

Answer: Thanks for the suggestion. When patients with ASD are taken into the dental unit, we try to treat them but it is not always possible. We tried to treat class-1 caries in all patients both in RDE and in SADE.  In the first phase in RDE the patient was seated on the dental chair and through the dental turbine the class-1 caries was removed and the filling was made with glass ionomer restorative material.

In the second phase, the patient was transferred to the SADE with soft lighting. The patient was then seated on the dental chair while a screen was projecting movies, cartoons or advertisements (already known by the subject and previously provided by the family). By using a  sponge-coated dental turbine drill to minimize the noise, the class-1 caries was removed and, once removed, the resulting cavity was filled with glass ionomer restorative material.

These details have been added in Materials and methods sections (2.2. Procedures: Environment), moreover a graph showing the study design have been also added (see figure 1).

RDE-11 patients and SADE- 34 patients were treated; the data on this context was missing.

Answer: Thanks for the suggestion. Please see the answer above.

I would be interested in seeing the data collection form as a supplemental file.

Answer: Thanks for the suggestion. The data of our study were extracted from the general form filled out for each patients treated in dental unit of Oasi Research Institute. We attached a copy of this form (modulo.pdf).

Discussion only based on RDE and SADE if authors could able to state the treatments, then it will help establish further prospects.

Answer: Thanks for the suggestion. When patients with ASD are taken into the dental unit, we try to treat them but it is not always possible. We tried to treat class-1 caries in all patients both in RDE and in SADE.  In the first phase in RDE the patient was seated on the dental chair and through the dental turbine the class-1 caries was removed and the filling was made with glass ionomer restorative material.

In the second phase, the patient was transferred to the SADE with soft lighting. The patient was then seated on the dental chair while a screen was projecting movies, cartoons or advertisements (already known by the subject and previously provided by the family). By using a  sponge-coated dental turbine drill to minimize the noise, the class-1 caries was removed and, once removed, the resulting cavity was filled with glass ionomer restorative material.

These details have been added in Materials and methods sections (2.2. Procedures: Environment), moreover a graph showing the study design have been also added (see figure 1).

The present form of discussion was not really focused on the object of the study. I also recommend the authors draw a flow chart that helps understand the study design.

Answer: Thanks for the suggestion. We added a flow chart (figure 1) as requested.

The below queries also not been clearly stated:

Methodology: there is confusion in subject selection kindly rephrase the statement.

Answer: Thanks for the suggestion. We changed the statement accordingly in the Methods sections.

  1. Need explanation on sample size. "Since dental decay, at least two teeth was a mandatory inclusion criterion, five subjects were excluded from the study; thus, the final sample consisted of 50 patients, 14 females, and 36 males." Comprehensive details are required on inclusion and exclusion criteria.

Answer: Thanks for the suggestion. We clarified the inclusion and exclusion criteria in Materials and Methods section (please see 2.1. Sample).

  1.  

Not very clear. Use the below citations for comparison with your findings.:

Fux-Noy A, Zohar M, Herzog K, Shmueli A, Halperson E, Moskovitz M, Ram D. The effect of the waiting room's environment on level of anxiety experienced by children prior to dental treatment: a case control study. BMC Oral Health. 2019 Dec 30;19(1):294.

Moana-Filho EJ, Alonso AA, Kapos FP, Leon-Salazar V, Durand SH, Hodges JS, Nixdorf DR. Multifactorial assessment of measurement errors affecting intraoral quantitative sensory testing reliability. Scand J Pain. 2017 Jul;16:93-98.

Panda A, Garg I, Shah M. Children's preferences concerning ambiance of dental waiting rooms. Eur Arch Paediatr Dent. 2015 Feb;16(1):27-33.

Cermak SA, Stein Duker LI, Williams ME, Dawson ME, Lane CJ, Polido JC. Sensory Adapted Dental Environments to Enhance Oral Care for Children with Autism Spectrum Disorders: A Randomized Controlled Pilot Study. J Autism Dev Disord. 2015;45(9):2876-2888. doi:10.1007/s10803-015-2450-5

Answer: Many thanks for your valued suggestions. All the study suggested are cited to support our results.

Reviewer 2 Report

Dear authors, thank you very much for the noticeable adaptations and the consistent implementation of the suggested modifications. From my point of view, this has added value to their wonderful work. Thank you very much for your engagement for people with an autism spectrum disorder and their access to the dental setting in terms of the specificities of the group of people.

Author Response

Dear Reviewers,

I would like to thank you for your valued comments and suggestions to the article. As requested, we made all the necessary changes in our manuscript to address your concerns which you can find detailed described below. The main changes are written in red in the manuscript. Thus, according to the changes made in the revised manuscript and the responses provided below, I hope you will consider the manuscript suitable for publication. If there are any further questions, please feel free to contact me.

Sincerely,

Luigi Vetri,

ORCID iD 0000-0002-0121-3396

Oasi Research Institute-IRCCS, Troina, Italy

Address: Via Conte Ruggero 73 - 94018 TROINA (EN);

Phone: +390935936809

Email: [email protected][email protected]

Reviewer 2

Dear authors, thank you very much for the noticeable adaptations and the consistent implementation of the suggested modifications. From my point of view, this has added value to their wonderful work. Thank you very much for your engagement for people with an autism spectrum disorder and their access to the dental l è+setting in terms of the specificities of the group of people.

Answer: Many thanks for your valued suggestions.

Round 2

Reviewer 1 Report

Authors addressed all the queries.

Author Response

Dear Reviewer,

I would like to thank you for your valued comments and suggestions to the article. 

Sincerely,

Luigi Vetri,

This manuscript is a resubmission of an earlier submission. The following is a list of the peer review reports and author responses from that submission.

Round 1

Reviewer 1 Report

Dear authors 

I read the paper with great interest.

what is the need for the study? not clear.

with the same context, there are studies already published, how your contribution will add value to published literature.

Add the blow references to an introduction:

Cermak SA, Stein Duker LI, Williams ME, Lane CJ, Dawson ME, Borreson AE, Polido JC. Feasibility of a sensory-adapted dental environment for children with autism. Am J Occup Ther. 2015 May-Jun;69(3):6903220020p1-10.

Ismail AF, Tengku Azmi TMA, Malek WMSWA, Mallineni SK. The effect of multisensory-adapted dental environment on children's behavior toward dental treatment: A systematic review. J Indian Soc Pedod Prev Dent. 2021 Jan-Mar;39(1):2-8.

Methodology: there is confusion in subject selection kindly rephrase the statement.

Need explanation on sample size.

"Since dental decay, at least two teeth was a mandatory inclusion criterion, five subjects were excluded from the study; thus, the final sample consisted of 50 patients, 14 females, and 36 males."

Comprehensive details are required on inclusion and exclusion criteria.

Results: 

Need to improve results section.

Shift demographic data to results.

Need an improvement by means of minute details of the study findings.

Discussion:

Not very clear 

Use the below citations:

Fux-Noy A, Zohar M, Herzog K, Shmueli A, Halperson E, Moskovitz M, Ram D. The effect of the waiting room's environment on level of anxiety experienced by children prior to dental treatment: a case control study. BMC Oral Health. 2019 Dec 30;19(1):294. 

Moana-Filho EJ, Alonso AA, Kapos FP, Leon-Salazar V, Durand SH, Hodges JS, Nixdorf DR. Multifactorial assessment of measurement errors affecting intraoral quantitative sensory testing reliability. Scand J Pain. 2017 Jul;16:93-98. 

Panda A, Garg I, Shah M. Children's preferences concerning ambiance of dental waiting rooms. Eur Arch Paediatr Dent. 2015 Feb;16(1):27-33.

Cermak SA, Stein Duker LI, Williams ME, Dawson ME, Lane CJ, Polido JC. Sensory Adapted Dental Environments to Enhance Oral Care for Children with Autism Spectrum Disorders: A Randomized Controlled Pilot Study. J Autism Dev Disord. 2015;45(9):2876-2888. doi:10.1007/s10803-015-2450-5

It is very difficult to find studies on autism subjects 

but there are a couple of studies focused on  and used for discussion 

Need a lot of improvement.

Conclusions are objective-based.

there are a couple of language errors, kindly address teh typo errors.

Overall a good attempt.

Author Response

Dear Reviewers,

I would like to thank you for your valued comments and suggestions to the article. As you requested, we made all the necessary changes in our manuscript to address your concerns and we detailed below how the points raised have been accommodated. The main changes are written in red in the text of the manuscript. From the changes made in the revised manuscript and responses provided below, I hope you are convinced that we have adequately addressed the reviewer’s concerns and made the paper better. If there are any further questions, please feel free to let me know.

Sincerely,

Luigi Vetri

Reviewer 1

Dear authors 

I read the paper with great interest.

what is the need for the study? not clear.

Answer: Many thanks for your valued suggestions. The need of the study is to reaffirm that sensory dysregulation in children with ASD is a crucial factor influencing the successful outcome of oral treatment.

with the same context, there are studies already published, how your contribution will add value to published literature.

Add the blow references to an introduction:

Cermak SA, Stein Duker LI, Williams ME, Lane CJ, Dawson ME, Borreson AE, Polido JC. Feasibility of a sensory-adapted dental environment for children with autism. Am J Occup Ther. 2015 May-Jun;69(3):6903220020p1-10.

Ismail AF, Tengku Azmi TMA, Malek WMSWA, Mallineni SK. The effect of multisensory-adapted dental environment on children's behavior toward dental treatment: A systematic review. J Indian Soc Pedod Prev Dent. 2021 Jan-Mar;39(1):2-8.

Answer: Thanks for the suggestion. We added your suggested studies in the introduction. We hope that, although it is similar to other existing studies, our study can lead to larger sample studies in future that will be able to determine which children will best benefit from the SADE intervention.

Methodology: there is confusion in subject selection kindly rephrase the statement.

Need explanation on sample size. "Since dental decay, at least two teeth was a mandatory inclusion criterion, five subjects were excluded from the study; thus, the final sample consisted of 50 patients, 14 females, and 36 males." Comprehensive details are required on inclusion and exclusion criteria.

Answer: Thanks for the suggestion. We reformulated the Materials and Methods section and better defined the inclusion and exclusion criteria.

Results:  Need to improve results section. Shift demographic data to results. Need an improvement by means of minute details of the study findings.

Answer: Thanks for the suggestion. We shifted the demographic data in the results section and improved the results analysing more deeply our data and by differentiating our results for sex.

Discussion:

Not very clear. Use the below citations:

Fux-Noy A, Zohar M, Herzog K, Shmueli A, Halperson E, Moskovitz M, Ram D. The effect of the waiting room's environment on level of anxiety experienced by children prior to dental treatment: a case control study. BMC Oral Health. 2019 Dec 30;19(1):294. 

Moana-Filho EJ, Alonso AA, Kapos FP, Leon-Salazar V, Durand SH, Hodges JS, Nixdorf DR. Multifactorial assessment of measurement errors affecting intraoral quantitative sensory testing reliability. Scand J Pain. 2017 Jul;16:93-98. 

Panda A, Garg I, Shah M. Children's preferences concerning ambiance of dental waiting rooms. Eur Arch Paediatr Dent. 2015 Feb;16(1):27-33.

Cermak SA, Stein Duker LI, Williams ME, Dawson ME, Lane CJ, Polido JC. Sensory Adapted Dental Environments to Enhance Oral Care for Children with Autism Spectrum Disorders: A Randomized Controlled Pilot Study. J Autism Dev Disord. 2015;45(9):2876-2888. doi:10.1007/s10803-015-2450-5

It is very difficult to find studies on autism subjects but there are a couple of studies focused on and used for discussion Need a lot of improvement.

Answer: Thanks for the suggestion. We added all your suggested articles. We used the suggested studies to expand and improve the discussion and to highlight the importance of the waiting room and not only of the operating room in dental care.

Conclusions are objective-based.

there are a couple of language errors, kindly address teh typo errors.

Answer: Thanks for the suggestion. We performed a linguistic revision.

Reviewer 2 Report

Thank you for the manuscript, which is overall suitable for the readership of the journal "children". However, it is only conditionally suitable for the selected special issue. Justification: The manuscript does not describe any significant progress and does not deal with a disease of the teeth/tooth structure in childhood or adolescence. It does emphasize the importance of the psychological aspects of dental treatment for individuals with autism spectrum disorder. A current and important topic is considered. Fundamentally, the importance of dental health as a component of general health is considered, especially in light of the challenges of managing individuals with autism spectrum disorder. The manuscript highlights an intervention-based approach, adapting spatial design and sensory input into dental treatment. The authors are to be fundamentally congratulated for this work, although unfortunately the manuscript has craft deficiencies and needs extensive revision. Below are some considerations so that the manuscript gains scientific value.

1.) Title: Appropriate - no adjustment necessary.

2.) Introduction: The introduction is overall understandable, but in the context of the length of the whole manuscript too long.

3.) Materials: Unfortunately, some important information and data is missing in this section. From my point of view this has to be adapted, especially to be able to classify and discuss the results accordingly. Why were all children/adolescents first treated according to the RDE and in a 2nd session in the SADE. Why was the group not randomized/split/cross-over? There is no information on the dental treatment content - which tooth was treated in each session. The statement that the inclusion criteria considered that the children had two carious sites is insufficient. It may represent a significant difference whether a deciduous tooth or a permanent tooth is treated. Furthermore, it must be assumed that if the children were 9-10years old, both dentitions were present. Also, in terms of the preventive concept, it is important to know which tooth required treatment. There is a lack of data on children's self-efficacy regarding oral hygiene. Which program was used for statistical analysis.

There is no information on the dentist. Was this always the same person, did they have experience in dealing with individuals with ASD, how much professional experience? Where were the treatments performed? Why 3 months apart?

It must also be questioned why these results of a study from 2014 are only now being published - these could possibly already be considered almost outdated.

4.) Results: This section is unfortunately very short and cannot be accepted in this form. The presentation of results covers only one aspect. Moreover, the figure is very general and does not even show the n (absolute or percentage). Furthermore, a detailed description of the study group (descriptive) is missing. No tooth-related parameters are presented - dmft/DMFT values, no clinical parameters, no description of how long the treatment time was (e.g. in RDE or SADE). On the basis of this very reduced presentation of results, to conclude on the positive effect of SADE seems almost overdrawn. Moreover, it must be noted that even in SADE, one third of the study group was still not successfully treated. It is not described whether the 11 successful treatments with RDE are also included in the 34 of SADE. That is, whether there were also study participants who were successfully treated with both RDE and SADE.

These or comparable results must be provided in order to make a scientific statement.

5.) Discussion: This section is very one-dimensional and could be much more forward-looking.

It is made clear why the chosen setting, the contents of the SADE or the individual adaptations were made, but it is not presented concretely enough why and how this can now be incorporated in clinical everyday life. Which prerequisites are necessary - e.g. training of the dental team. Far away, due to the lack of detailed presentation of results, no obtained results are discussed. However, this must be done. E.g. - If it is evident that boys were more often treatable with both methods than girls - what could be the reasons for this observation.

Other editorial comment:

1.) It should be reconsidered whether some of the literature should not be updated. Except for one literature, the others are from 1988 - 2016.

Author Response

Dear Reviewers,

I would like to thank you for your valued comments and suggestions to the article. As you requested, we made all the necessary changes in our manuscript to address your concerns and we detailed below how the points raised have been accommodated. The main changes are written in red in the text of the manuscript. From the changes made in the revised manuscript and responses provided below, I hope you are convinced that we have adequately addressed the reviewer’s concerns and made the paper better. If there are any further questions, please feel free to let me know.

Sincerely,

Luigi Vetri

Reviewer 2

Thank you for the manuscript, which is overall suitable for the readership of the journal "children". However, it is only conditionally suitable for the selected special issue. Justification: The manuscript does not describe any significant progress and does not deal with a disease of the teeth/tooth structure in childhood or adolescence. It does emphasize the importance of the psychological aspects of dental treatment for individuals with autism spectrum disorder. A current and important topic is considered. Fundamentally, the importance of dental health as a component of general health is considered, especially in light of the challenges of managing individuals with autism spectrum disorder. The manuscript highlights an intervention-based approach, adapting spatial design and sensory input into dental treatment. The authors are to be fundamentally congratulated for this work, although unfortunately the manuscript has craft deficiencies and needs extensive revision. Below are some considerations so that the manuscript gains scientific value.

1.) Title: Appropriate - no adjustment necessary.

2.) Introduction: The introduction is overall understandable, but in the context of the length of the whole manuscript too long.

Answer: Many thanks for your valued suggestions. We remodulated all parts of the manuscript and we believe that the article parts are now well balanced.

3.) Materials: Unfortunately, some important information and data is missing in this section. From my point of view this has to be adapted, especially to be able to classify and discuss the results accordingly. Why were all children/adolescents first treated according to the RDE and in a 2nd session in the SADE. Why was the group not randomized/split/cross-over?

Answer: Thanks for the suggestion. The aim of the study is to evaluate if changes of dental environmental influence the dental treatment in a sample of children with ASD. Therefore, all patients of the same sample of children with ASD were subjected to two dental treatments, the first in a regular dental environment (RDE) and the second in a sensory-adapted dental en-vironment (SADE).

There is no information on the dental treatment content - which tooth was treated in each session. The statement that the inclusion criteria considered that the children had two carious sites is insufficient. It may represent a significant difference whether a deciduous tooth or a permanent tooth is treated. Furthermore, it must be assumed that if the children were 9-10years old, both dentitions were present. Also, in terms of the preventive concept, it is important to know which tooth required treatment. There is a lack of data on children's self-efficacy regarding oral hygiene. Which program was used for statistical analysis. There is no information on the dentist. Was this always the same person, did they have experience in dealing with individuals with ASD, how much professional experience? Where were the treatments performed? Why 3 months apart?

Answer: Thanks for the suggestion. We added all information requested. Inclusion and exclusion criteria were better defined. All tooths treated were permanent with caries in Class I. Statistical procedures were performed with SPSS - version 26.0.0.0. Moreover, we specified that dentist was always the same and that the dental treatments of 50 patients have been scheduled one per day, therefore the average time elapsed between the RDE and the SADE treatment was two months.

It must also be questioned why these results of a study from 2014 are only now being published - these could possibly already be considered almost outdated.

Answer: Thanks for the suggestion. Oasi Research Institute is a primary care center therefore some research requires some time to complete. Moreover, the recent COVID-19 pandemic has almost completely frozen our activity. Nevertheless, we do not believe that our data can be influenced by the passage of time.

4.) Results: This section is unfortunately very short and cannot be accepted in this form. The presentation of results covers only one aspect. Moreover, the figure is very general and does not even show the n (absolute or percentage). Furthermore, a detailed description of the study group (descriptive) is missing. No tooth-related parameters are presented - dmft/DMFT values, no clinical parameters, no description of how long the treatment time was (e.g. in RDE or SADE). On the basis of this very reduced presentation of results, to conclude on the positive effect of SADE seems almost overdrawn. Moreover, it must be noted that even in SADE, one third of the study group was still not successfully treated. It is not described whether the 11 successful treatments with RDE are also included in the 34 of SADE. That is, whether there were also study participants who were successfully treated with both RDE and SADE. These or comparable results must be provided in order to make a scientific statement.

Answer: Thanks for the suggestion. The result section has been improved; the n value has been added. All patients (11) successfully treated in RDE were at a later date treated in SADE. A table showing the sex differences has been added and the graph was reformulated.

5.) Discussion: This section is very one-dimensional and could be much more forward-looking.

It is made clear why the chosen setting, the contents of the SADE or the individual adaptations were made, but it is not presented concretely enough why and how this can now be incorporated in clinical everyday life. Which prerequisites are necessary - e.g. training of the dental team. Far away, due to the lack of detailed presentation of results, no obtained results are discussed. However, this must be done. E.g. - If it is evident that boys were more often treatable with both methods than girls - what could be the reasons for this observation.

Answer: Thanks for the suggestion. We discuss more deeply in this section our results, and we suggest the possible explanations for sex difference present in our study.

Other editorial comment:

1.) It should be reconsidered whether some of the literature should not be updated. Except for one literature, the others are from 1988 - 2016.

Answer: Thanks for the suggestion. We updated the literature adding several recent citations.

Round 2

Reviewer 1 Report

Dear author

Thank you for inputs.

The research problem should be better described in the Introduction section.
- The very long paragraphs (with many sentences) in the Discussion section mainly make the article very difficult to read and understand. Please review this.
- The authors do not clearly mention the limitations of the study. This should be reviewed.
- The Results section is poor. Authors should provide more details.
- The Conclusion section should be rewritten. 
- Highlights  the updated reference list.

Author Response

Dear Reviewer,

I would like to thank you for your valued comments and suggestions to the article. As you requested, we made all the necessary changes in our manuscript to address your concerns and we detailed below how the points raised have been accommodated. The main changes are written in red in the text of the manuscript. From the changes made in the revised manuscript and responses provided below, I hope you are convinced that we have adequately addressed your concerns and made the paper better. If there are any further questions, please feel free to let me know.

Sincerely

Luigi Vetri

-The research problem should be better described in the Introduction section.

Answer: Many thanks for your valued suggestions. The topic of our research has been better analysed in the introduction section (please see lines 83-90)

- The very long paragraphs (with many sentences) in the Discussion section mainly make the article very difficult to read and understand. Please review this.

Answer: Thanks for the suggestion. All the text of the manuscript has been linguistically revised.

- The authors do not clearly mention the limitations of the study. This should be reviewed.

Answer: Thanks for the suggestion. The limitations of the study have been highlighted in the discussion section (please see lines 176-179)

- The Results section is poor. Authors should provide more details.

Answer: Thanks for the suggestion. The result section has been revised and more details have been added. (135-145)

- The Conclusion section should be rewritten. 

Answer: Thanks for the suggestion. The conclusions have been rewritten (please see lines 221-224)

- Highlights  the updated reference list.

Answer: Thanks for the suggestion. We highlighted the added references.

Reviewer 2 Report

Dear authors, thank you very much for the noticeable adaptations and the consistent implementation of the suggested modifications. From my point of view, this has added value to their wonderful work. Thus, the manuscript is very suitable for publication. Thank you very much for your engagement for people with an autism spectrum disorder and their access to the dental setting in terms of the specificities of the group of people.

Author Response

Dear Reviewer,

I would like to thank you for your valued comments and suggestions to the article. As you requested, we made all the necessary changes in our manuscript to address your concerns and we detailed below how the points raised have been accommodated. The main changes are written in red in the text of the manuscript. From the changes made in the revised manuscript and responses provided below, I hope you are convinced that we have adequately addressed your concerns and made the paper better. If there are any further questions, please feel free to let me know.

Sincerely,

Luigi Vetri

Dear authors, thank you very much for the noticeable adaptations and the consistent implementation of the suggested modifications. From my point of view, this has added value to their wonderful work. Thus, the manuscript is very suitable for publication. Thank you very much for your engagement for people with an autism spectrum disorder and their access to the dental setting in terms of the specificities of the group of people.

Answer: Dear reviewer, many thanks for your valued suggestions and for your contribution to the improvement of our manuscript.